# Production of Indole and Indole-Related Compounds by the Intestinal Microbiota and Consequences for the Host: The Good, the Bad, and the Ugly

**DOI:** 10.3390/microorganisms10050930

**Published:** 2022-04-28

**Authors:** Naouel Tennoune, Mireille Andriamihaja, François Blachier

**Affiliations:** Université Paris-Saclay, AgroParisTech, Inrae, UMR PNCA, 75005 Paris, France; mireille.andriamihaja@agroparistech.fr (M.A.); francois.blachier@agroparistech.fr (F.B.)

**Keywords:** indole, isatin, indoxyl sulfate, gut microbiota, intestine, kidney, brain

## Abstract

The intestinal microbiota metabolic activity towards the available substrates generates myriad bacterial metabolites that may accumulate in the luminal fluid. Among them, indole and indole-related compounds are produced by specific bacterial species from tryptophan. Although indole-related compounds are, first, involved in intestinal microbial community communication, these molecules are also active on the intestinal mucosa, exerting generally beneficial effects in different experimental situations. After absorption, indole is partly metabolized in the liver into the co-metabolite indoxyl sulfate. Although some anti-inflammatory actions of indole on liver cells have been shown, indoxyl sulfate is a well-known uremic toxin that aggravates chronic kidney disease, through deleterious effects on kidney cells. Indoxyl sulfate is also known to provoke endothelial dysfunction. Regarding the central nervous system, emerging research indicates that indole at excessive concentrations displays a negative impact on emotional behavior. The indole-derived co-metabolite isatin appears, in pre-clinical studies, to accumulate in the brain, modulating brain function either positively or negatively, depending on the doses used. Oxindole, a bacterial metabolite that enters the brain, has shown deleterious effects on the central nervous system in experimental studies. Lastly, recent studies performed with indoxyl sulfate report either beneficial or deleterious effects depending once again on the dose used, with missing information on the physiological concentrations that are reaching the central nervous system. Any intervention aiming at modulating indole and indole-related compound concentrations in the biological fluids should crucially take into account the dual effects of these compounds according to the host tissues considered.

## 1. Introduction

The colonic epithelial cells are polarized cells that face, on their luminal side, a complex mixture of microorganisms and molecules. These molecules include several bacterial components and myriad bacterial metabolites that are metabolic intermediaries and final products derived from the colonic bacterial metabolic activity that may accumulate in the intestinal luminal content. These bacterial metabolites have attracted increasing interest due to their capacity to interact with different aspects of colonic epithelial cell metabolism and physiology [1], and their potential implication in the development or modulation of pathological processes that occur in the colonic and rectal mucosae, notably chronic inflammatory bowel disease. These diseases have been associated with changes in the colonic bacterial metabolite profiles [2], raising the possibility of causal links between such changes and the modulation of the risk of colorectal inflammation. Among bacterial metabolites, those derived from various amino acids released from dietary and endogenous proteins present in the intestinal content represent a family of molecules, whose concentrations can vary according to many parameters, notably and interestingly including dietary habits [3]. 

Several amino acid-derived bacterial metabolites can be absorbed through the intestinal epithelial cells and metabolized by the host tissues, giving rise to the so-called co-metabolites, which thus result from the joint activities of the intestinal microbiota and host cells. The effects of co-metabolites on the different host peripheral tissues in different physiological or pathological situations represent an important issue to establish the overall consequences on the host of the production of bacterial metabolites.

In such a context, the present paper aims to provide the state of the art regarding the production from L-tryptophan of indole and indole-related compounds by the intestinal microbiota, the effects of indolic compounds on the intestinal mucosa, the metabolism of indole by the host, and the effects of indole and co-metabolites on peripheral organs. For the sake of clarity and concision, this review does not concern the effects on the host of other tryptophan-derived bacterial metabolites such as tryptamine, an inducer for the release of 5-hydroxytryptamine (serotonin) by enteroendocrine cells; this latter compound is able to stimulate gastrointestinal motility by acting on the neurons of the enteric nervous system, as detailed in recent reviews on that topic [4,5,6]. 

## 2. The Amino Acid L-tryptophan Is Used by the Intestinal Microbiota to Produce Indole and Indole-Related Compounds That Are Recovered in the Large Intestine Fluid

L-tryptophan is one of the nine indispensable amino acids that must be provided in the diet to meet the body’s requirements [7]. L-tryptophan, in addition to its role as a precursor for protein synthesis in host tissues, is also a precursor for numerous compounds with biological activities, including the neurotransmitters serotonin and tryptamine, the hormone melatonin, and the vitamin-like compounds niacin and nicotinic acid [8]. In humans, the tryptophan requirement is 4.0 mg/kg body weight/day. The mean amount of tryptophan provided from all sources is 0.91 g/day, thus representing a usual consumption equal to 13.0 mg/kg body weight/day; thus, this value is well above the requirement [9]. In addition to being used by the different organs and tissues of the host body, a minor part of available tryptophan, approximately 5%, is metabolized by the gut microbes [5]. Exogenous tryptophan used by the gut microbes mainly originates from the undigested, or not fully digested, proteins, which are transferred from the small to the large intestine [10]. In fact, based on a regular Western diet, 6–12 g of proteins and peptides from both dietary and endogenous origin escape digestion in the small intestine, thus reaching the colonic lumen [11,12,13,14] (Figure 1). This quantity of nitrogenous materials can be usefully compared to the dietary protein consumption in Western countries, which averages 85 g per day [15,16]. 

Bacterial proteases and peptidases release amino acids, including tryptophan, from available dietary and endogenous luminal protein, and since amino acids are little absorbed by the colonic epithelial cells [17], they are mainly used by the large intestine microbes in the course of their metabolic activity. 

Intestinal bacteria convert tryptophan mainly into indole through the action of the bacterial enzyme tryptophanase; the latter is induced by tryptophan itself [18]. Of note, in mammals, indole originates exclusively from bacterial metabolic activity since host cells do not have the metabolic capacity for the production of this compound [19]. Indole is synthesized from various Gram-positive and Gram-negative bacterial species, including *Escherichia coli*, *Proteus Vulgaris*, *Clostridium* spp., and *Bacteroides* spp. [6,19,20]. Tryptophan can also be converted directly or indirectly by the intestinal microbiota into several indole-related compounds, including indole-3-pyruvate, indole-3-lactate, indole-3-propionate, indole-3-acetamide, indole-3-acrylate, indole acetaldehyde, indole-3-acetate, indole-3-aldehyde, 3-methyl-indole (skatole), and indole-3-acetaldehyde [5,18,21,22,23,24]. However, the precise metabolic pathways involved in the stepwise conversion of tryptophan into these minor indolic compounds by the intestinal microbiota in the large intestine remain unclear. Regarding the bacterial species involved in their production, indole-3-propionate has been shown to be produced by *Clostridium sporogenes* [25]. Skatole is known to be produced by *Clostridium* spp. and *Bacteroides* spp. [6,26,27]. Other indolic compounds are produced by *Bacteroides* species, *Peptostreptococcus* spp. *Lactobacillus* spp., and *Bifidobacterium* spp. [6,28]. 

The fecal indole content has been measured in volunteers, but huge inter-individual differences exist between volunteers, ranging from 0.30 to 6.64 millimolar concentrations [29]. Such a large range is presumably due to differences in intestinal microbiota composition between volunteers and to different levels of dietary protein consumption [30]. Incidentally, since indole appears to be absorbed through the colonic epithelium [31], the concentration of indole in the colon is likely higher than that recorded in the feces, with concentrations higher in the distal large intestine than in the more proximal parts [32].

Fecal skatole concentration in healthy individuals averages 0.04 mM [33], but the concentration in the colonic content is again presumably higher since skatole is absorbed through the gut epithelium [34], and then released in the circulation [35]. In the pig model, skatole colonic concentration averages 0.23 mM [36]. Skatole concentration increases in the large intestine after a high-meat diet, or when luminal fermentation increases as a result of longer intestinal stasis [33]. In rodents, indole-3-propionate is found only in blood recovered from animals with intestinal microbiota, but not in blood originating from germ-free animals [37], thus reinforcing the view that this metabolite originates from the intestinal microbiota’s metabolic activity. 

## 3. Indole and Indolic Compounds Are Involved in Intestinal Microbial Community Communication

Among amino acid-derived bacterial metabolites, indole has attracted growing interest since this compound is involved in bacterial physiology and metabolism in relationship with antibiotic resistance, virulence factors, sporulation, and biofilm formation [19,38,39,40]. Indeed, indole is a bacterial signal involved in the communication between bacteria within the same species and between bacteria of different species. It acts as an inter-and intra-cellular signal in bacterial ecosystems (Figure 1). Indole diminishes the related virulence of *L. monocytogenes* by reducing cell motility and aggregation [41]. Indole influences host cell invasion by non-indole-producing species, such as *Salmonella enterica* and *P. aeruginosa*, and the fungal species *Candida albicans* [42] In addition, indole reduces *E. coli* motility [43]. In another study, indole was shown to display bacteriostatic effects on lactic acid bacteria, while affecting their survival [44]. Of note, indole mitigates cytotoxicity by *Klebsellia spp*. by suppressing toxin production [45].

## 4. Indole and Indolic Compounds Exert Beneficial Effects on the Intestinal Mucosa

Several indolic compounds, including indole-3-acetate, indole-3-propionate, indole-3-aldehyde, indole-3-acetaldehyde, and indole acrylate, bind to the aryl hydro carbon receptor [4], which is present in different cell types of the host, including cells present in the intestinal mucosa, notably intestinal epithelial cells and immune cells [46]. The binding of indolic compounds to AhR participates in the maintenance of the intestinal mucosa homeostasis by acting on the control of the intestinal epithelium renewal, its barrier function, and the activity of several intestinal immune cell types [47]. AhR can also be activated by dietary compounds [48], but bacterial metabolites appear to play a preponderant role in the AhR activation [49]. Only a few commensal bacteria have been identified as able to produce AhR ligands, such as *Peptostreptococcus russellii* [28] and *Lactobacillus* spp. [50]. 

Among the indolic compounds, indole-3-propionate, when given by the oral route, exerts beneficial effects on the intestinal barrier function when the latter is experimentally altered in rodent models, such as via radiation injury [51], or rodents fed a high-fat diet [52]. In addition, indole-3-propionate reduces intestinal permeability and inflammation in a rodent model [53]. In such an experimental context, it is of interest to note that circulating indole-3-propionate acid is reduced in patients suffering from ulcerative colitis when compared with their healthy counterparts, whereas an increased level of this bacterial metabolite is associated with remission [54].

In a model of experimental colitis induced in mice, other indolic compounds given by the oral route, namely indole-3-pyruvate, indole-3-aldehyde, and indole-3-ethanol, protect against increased intestinal permeability observed in this model [55]. Indole itself, when given by the oral route, decreases mucosal inflammation and injury in a model of enteropathy in rodents [56]. In mice, microbiota-derived indole-3-aldehyde contributes to aryl hydrocarbon receptor-dependent IL-22 gene transcription, allowing the survival of mixed microbial communities, while providing colonization resistance to the fungal species *C. albicans*, and protecting the mucosa from inflammation [50]. Indole-3-acrylate diminishes in mice intestinal inflammation and upregulates Mucin 2 gene expression [28].

Protective effects of microbiota-derived aryl hydrocarbon receptor agonists on the intestinal mucosa have been suggested by both experimental and clinical data, with presumed consequences for the risk of metabolic syndrome [57]. In this latter study, metabolic syndrome was found to be associated with an impaired capacity of the gut microbiota to produce from tryptophan aryl hydrocarbon agonist receptors; this impaired capacity is paralleled by increased gut permeability and decreased secretion of GLP-1 from enteroendocrine cells [57]. 

Regarding the effects of indole on intestinal epithelial cells, exposure of human enterocytes to indole, but not to other indole-like compounds, increases the expression of genes involved in the intestinal epithelial barrier function and mucin production [58]. Interestingly, these effects were paralleled by an effect of indole on the expression of different cytokines, with decreased expression of the pro-inflammatory IL-8, and increased expression of the regulatory cytokine IL-10. However, in this study, the expression of several genes linked to inflammation was also found to be increased [58], making the indole effect on the intestinal epithelial cells more complicated than it appears at first sight.

Moreover, oral administration of an indole-containing capsule to rodents results in an increased expression in colonocytes of genes coding for tight junction proteins between epithelial cells [59]. Following these results, indole was found to increase transepithelial resistance in in vitro experiments using colonocyte monolayers [60], thus reinforcing the view that indole ameliorates the basal barrier function. Thus, indole and several indolic compounds exert beneficial effects on the intestinal mucosa in different situations (Figure 1). Overall, the protective effects of bacterial metabolites derived from tryptophan on intestinal mucosa may contribute to the beneficial effects of tryptophan supplementation observed in animal models of colitis [61,62].

However, surprisingly, indole used at a 2.5 millimolar concentration affects the respiration of colonocytes by diminishing mitochondrial oxygen consumption [60], and thus mitochondrial ATP production. This last effect was accompanied by transient oxidative stress in colonocytes, which was followed by an increase in the expression of antioxidant enzymes, presumably as an adaptive process against the deleterious effect of excessive indole exposure. 

Little is known about the effects of skatole on intestinal cells, but high concentrations of skatole induce cell death in human colonocytes [63].

In in vitro experiments performed with immortalized and primary mouse colonic enteroendocrine L cells, indole modulates the secretion of glucagon-like peptide-1 (GLP-1) [64]. Since GLP-1 slows down gastric emptying, stimulates insulin secretion by pancreatic beta cells, and diminishes appetite [65], it would be of major interest to study in vivo the effects of indole on these physiological parameters.

## 5. After Absorption, Indole Is Partly Metabolized into Indoxyl Sulfate in the Liver

Indole, after absorption in the portal vein, has been shown to exert some anti-inflammatory effects on liver cells (Figure 1). In a rodent model, indole reduces the production of pro-inflammatory mediators by the liver [66]. In a model of obese mice, indole reduces hepatic damage and the associated inflammatory response [67]. The indolic compound indole-3-acetic acid alleviates, in mice, the high-fat diet-induced hepatotoxicity [68]. This latter bacterial metabolite reduces the expression of fatty acid synthase in hepatocytes [69]. 

The indole that reaches the liver cells is partly metabolized by several cytochrome (CYP) family enzymes, including, notably, CYP2E1 [70]. Several molecules are produced from indole, with indoxyl sulfate being the main co-metabolite produced [71] (Figure 1). As can be anticipated, subjects who consumed a high-protein diet showed overall greater indoxyl sulfate urinary excretion than those who consumed a low-protein diet [72]. In addition, in a randomized, parallel, double-blind trial in overweight volunteers, protein supplementation provoked an increased concentration of indoxyl sulfate in urine [73]. CYP enzymes are presumably involved in the conversion of indole to indoxyl, an intermediate in the synthesis of indoxyl sulfate. CYP2E1 represents the major enzymatic isoform responsible for the oxidation of indole to indoxyl [74]. CYP2E1 is detected in the colonic epithelium [75,76]. However, data obtained with human colonocytes of the HT-29 Glc-/+ line, chosen because these cells have retained metabolic features that are characteristic of healthy colonocytes (such as the capacity to oxidize butyrate and acetate [77,78], show that the capacity of these cells to convert indole to indoxyl sulfate is modest but measurable [60], and the vast majority of indoxyl sulfate is presumably produced within the liver (Figure 1). 

## 6. Indoxyl Sulfate Is a Recognized Uremic Toxin That Exerts Deleterious Effects on Tubular Kidney Cells and Accelerates Chronic Kidney Disease

Indoxyl sulfate after production in the liver is released in the peripheral blood and then excreted in urine [79]. Indoxyl sulfate concentrations in the blood can be increased from micromolar concentrations in healthy individuals, up to 1.1 millimolar in severe chronic kidney disease [80]. Indoxyl sulfate belongs to the family of uremic toxins that may accumulate in body fluids leading to the so-called uremic syndrome [81]. Indeed, indoxyl sulfate at excessive concentrations aggravates chronic kidney disease in patients [82]. In healthy individuals, indoxyl sulfate is almost entirely bound to proteins in the blood (approximately 93% of this co-metabolite is in bound form [83]). The main binding protein in the blood is albumin, with two binding sites for indoxyl sulfate [84]. Circulating IS in free form is then efficiently excreted in the urine by proximal tubular cells through basolateral organic anion transporters [85].

However, in patients with chronic kidney disease, only 85% of this co-metabolite is protein-bound [79], and thus a higher part of indoxyl sulfate is in free form. As kidney function declines, indoxyl sulfate total concentration increases in the blood and this elevation contributes to further progression of chronic kidney disease [86]. In a cohort study, it was found that blood indoxyl sulfate concentrations are higher in patients with chronic kidney failure progression than in stabilized patients [87]. Then, indoxyl sulfate concentration in blood has been proposed as an indicator of chronic kidney disease progression in dialyzed patients [88]. 

In experimental in vitro and in vivo studies, indoxyl sulfate has been shown to have deleterious effects on kidney cells when present in excess (Figure 1). This co-metabolite increases the expression of inflammation-associated genes in cultured proximal renal tubular cells [89]. This effect coincides with a capacity of indoxyl sulfate to increase the net production of reactive oxygen species in the proximal tubular cells [90,91], and oxidative stress in these cells [92]. In addition, this compound reduces the glutathione concentration in renal tubular cells [93]. Taking into account the central role of reduced glutathione in the process of excessive reactive oxygen species disposal [94], it is plausible that a reduced intracellular concentration of glutathione will render renal tubular cells even more vulnerable to oxidative stress. The fact that indoxyl sulfate also reduces the superoxide scavenging activity in the kidneys of normal and uremic rodents [95] contributes to the sensitivity of the kidney cells to oxidative stress. Indeed, superoxide is one of the reactive oxygen species that is deleterious to cells when its intracellular concentration exceeds a threshold value [96]. These effects are of major importance when considering that reactive oxygen species are elevated in renal tubular cells in the process of chronic kidney disease progression [97]. 

Indoxyl sulfate in excess is involved in renal fibrosis. Briefly, renal fibrosis results notably from excessive accumulation of extracellular matrix after renal insult [98]. Administration of indoxyl sulfate in experimental models of chronic kidney disease leads to glomerular sclerosis and interstitial fibrosis [99]. These effects of indoxyl sulfate can be explained by its capacity to promote the transformation of kidney fibroblasts into matrix-producing phenotype, thus increasing collagen deposition, a process that is linked to interstitial fibrosis [100]. This proposition is reinforced by the fact that indoxyl sulfate increases the expression of genes implicated in kidney fibrosis [101].

## 7. Indoxyl Sulfate Is Deleterious for Other Cell Types including Colonocytes and Endothelial Cells

Indoxyl sulfate may be toxic for colonocytes. The way of entry of this compound into colonocytes is most likely from the blood to the basolateral side of polarized cells. Indeed, as expected, the indoxyl sulfate concentration in the colonic content is in the low micromolar range [102]. At micromolar concentrations, indoxyl sulfate is cytotoxic for the intestinal epithelial cells, inducing oxidative stress and impairing cell migration [103]. At higher concentrations (millimolar), indoxyl sulfate reduces mitochondrial oxygen consumption in colonocytes, induces a transient increase in the reactive oxygen species’ net production, and rapidly but transiently increases the secretion of the pro-inflammatory IL-8 and, after a 48 h exposure, tumor necrosis factor-alpha (TNF-α) [60]. 

Finally, indoxyl sulfate can be included in the family of uremic endotheliotoxins, meaning that this compound induces endothelial dysfunction, one central element implicated in cardiovascular morbidity and mortality, taking into account that the risk of cardiovascular dysfunctions is associated with chronic kidney disease [104]. From in vivo and in vitro preclinical studies, indoxyl sulfate has been shown to promote both pro-thrombotic processes, notably through mechanisms involving the aryl hydrocarbon receptor [105,106,107] and pro-oxidant processes [108,109,110,111]. Clinical data obtained in patients with chronic kidney disease indicate that indoxyl sulfate is likely to represent one of the links between impaired renal function and adverse cardiovascular events, notably regarding hemostatic disorders [112] and thrombotic events [113]

## 8. Indole and Indolic Compounds Are Active on the Central Nervous System

There is emerging evidence of the influence of indole and indolic compounds on brain metabolism, physiology, and host behavior. 

### 8.1. Indole in Excess Shows Adverse Effects on Behavior

Chronic overproduction of indole in rats monocolonized with *E. coli* generating indole has been shown to enhance anxiety-like behavior and depression in these animals [114]. This latter study also found that giving intra-cecal indole to conventional rats activated a cerebral nucleus called the dorsal vagal complex. By comparing mice mono-associated with a non-indole-producing *E. coli* strain, or with an indole-producing *E. coli* strain, it was found that chronic high indole production by the intestinal microbiota increased the vulnerability to the adverse effects of chronic stress on overall emotional behavior [115]. In humans, a study has revealed that children with autism spectrum disorders reported lower levels of indole and increased levels of 3-methylindole in fecal samples in comparison to healthy children [116]. These two indoles have been linked to bacteria of the genus *Clostridium*, which are more prevalent in these patients [116]. A rise in plasma indole derived from intestinal microbiota metabolic activity is associated with hepatic encephalopathy, neuropsychiatric trouble caused by hepatic dysfunction, and changes in the individual’s state of consciousness, conduct, and personality [117]. Furthermore, the findings of the observational prospective NutriNet-Santé Study revealed a positive link between urine indole and indolic compounds’ concentrations and recurrent depressive symptoms. This correlation raises the hypothesis that the synthesis of these compounds by the gut microbiota in excessive amounts may play a role in the emergence of mood disorders in humans [118]. 

Some information on the effects of indolic compounds on the central nervous system are available. Isatin, oxindole, and indoxyl sulfate are the most studied indolic metabolites for their neuroactive effects (Figure 1).

### 8.2. Isatin Can Enter the Brain and Affect Behavior and Brain Function

Isatin (1H-indole-2,3-dione) is a co-metabolite produced by the host from indole [70]. Isatin is detected in different brain regions [119]. Systemic administration of isatin to rodents leads to an accumulation of this compound in the brain [116]. In addition, administration of indole in the large intestine results in an accumulation of isatin in the brain [116], thus suggesting that this compound is able to enter the brain by still unknown processes. The hippocampus and cerebellum have the largest concentrations of isatin in the rat brain, whereas the prefrontal cortex and brainstem have the lowest quantities [120]. The impact of isatin on rodent behavior and brain function, either beneficial or deleterious, appears to depend on the doses used [121,122]. Intraperitoneal treatment of isatin up to 20 mg/kg body weight promotes an increase in anxiety-like behavior in rats and mice [123]. Isatin affects anxiety-like behavior via acting on monoaminergic receptors. Furthermore, at 10 mg/kg, isatin reduces the anxiolytic effect of intracerebroventricular atrial natriuretic peptide (ANP) injection [124]. Because ANP has a role in the control of the hypothalamic-pituitary-adrenal (HPA) axis, it is plausible that isatin anxiogenic action would be mediated through this axis. Indeed, isatin (20 mg/kg) causes a 50% rise in plasma cortisol in rhesus monkeys [123]. Higher amounts of isatin, over 40 mg/kg when given intraperitoneally, result in a reduction in locomotor activity in the open field test and mobility in the forced swimming test in rats [124], indicating that isatin may exert a sedative effect.

Intraperitoneal injection of isatin (100 mg/kg) improved parkinsonian symptoms in a parkinsonian rat model induced by a 6-hydroxydopamine lesion [125]. Isatin intraperitoneal administration (100 mg/kg) increased dopamine concentration in the striatum of rats in the model of Parkinson’s disease induced by the Japanese encephalitis virus [126]. Isatin can inhibit, in a dose-dependent manner, the monoamine oxidase activities in extracts obtained from the rat brain [127]. It has been suggested that the effects of isatin, which is anxiogenic at low doses and sedative at high doses, would be related to the monoaminergic systems in the brain. Because of the affinity of isatin for ANP peptide receptors and their role in the stress axis, these receptors seem to represent one significant target in the mechanism of action of isatin on the central nervous system physiology and the anxiety-like behaviors.

### 8.3. Oxindole Can Enter the Brain and Exert Deleterious Effects on the Brain

The intraperitoneal injection of oxindole results in its accumulation in the brain [128], thus indicating that this compound may, like isatin, enter the brain by a still unknown process. Oral administration in rodents of neomycin, a broad-spectrum antibiotic, decreases the brain oxindole content [128], thus suggesting that oxindole is originating, at least partly, from the intestinal microbiota metabolic activity. Interestingly, oxindole was recently found in human fecal samples [129], thus confirming that the intestinal microbes are a source of this metabolite. Among the bacterial metabolites present in human stool, oxindole was found to be one of the dominant aryl hydrocarbon receptor activators [129]. In vitro experiments on rat hippocampus slices demonstrate that oxindole may interact with voltage-gated sodium channels, increasing the threshold for producing action potentials and therefore drastically reducing neuron excitability [130]. These results led several authors to propose that oxindole may share some characteristics with known neurodepressant compounds [128,130]. Of note, unlike isatin, oxindole has relatively low monoamine oxidase inhibitory activity [131].

### 8.4. Indoxyl Sulfate, Depending on the Dose Used, May Exert Both Beneficial and Deleterious Effects on Brain

Concerning indoxyl sulfate, this co-metabolite is detected in the mammalian brain [132]. Indoxyl sulfate is also detected in the cerebrospinal fluid of mice, and animals with no intestinal microbiota display a low level of this compound in this fluid when compared with normal mice hosting microbes [133]. By comparing volunteers suffering from depression with healthy participants, 22 urine metabolites were identified, and their abundances differed between the two groups of subjects. The Hamilton Depression Scale questionnaire score was used to measure the severity of depression. According to this investigation, the urine indoxyl sulfate concentration in individuals suffering from severe depression was lower than that in healthy counterparts [134]. In volunteers, serum indoxyl sulfate concentrations have been associated with psychic anxiety and the related functional magnetic resonance imaging-based neurological signature [135]. 

In a mouse model of experimental autoimmune encephalomyelitis, indoxyl sulfate delivered daily (10 mg/kg) intraperitoneally modulates astrocyte activity and exerts an anti-inflammatory action on the central nervous system via the aryl hydrocarbon receptor [136]. Conversely, when rats were exposed orally to indoxyl sulfate used at higher doses (100 and 200 mg/kg), impairment of spatial memory and reduced locomotor and exploratory activities were observed [137]. In this latter study, after indoxyl sulfate administration, this compound was mainly recovered in the brainstem. Another study found that a single intraperitoneal injection of indoxyl sulfate at an even greater dose (800 mg/kg) caused histological changes in the brain compatible with neuronal necrosis [138]. In this latter study, indoxyl sulfate, when used in a 15–60 micromolar range, induced radical oxygen species production in primary astrocytes, and cell death in hippocampal neurons [138]. At a 10-micromolar concentration, indoxyl sulfate induces apoptosis through oxidative stress in human astrocytes [139]. Unfortunately, little information is available on the physiological concentrations of indoxyl sulfate and of the other indole-related compounds that reach the central nervous system.

## 9. Conclusions and Perspectives

Indole and indole-related compounds are, first, involved in intestinal microbial community communication, regulating important aspects of bacterial physiology. These molecules are also known to be active on the intestinal mucosa, exerting overall beneficial effects in different experimental situations, notably in inflammatory situations. After absorption, indole is partly metabolized in the liver into the co-metabolite indoxyl sulfate. Indoxyl sulfate is well known to be one of the uremic toxins that can seriously aggravate chronic kidney disease, notably through deleterious effects on kidney cells. Regarding the central nervous system, emerging research indicates that indole at excessive concentrations displays a negative impact on emotional behavior in different experimental situations. From a limited number of experimental studies, the indolic compounds isatin, oxindole, and indoxyl sulfate also appear to affect the central nervous system activity.

Any intervention, either of dietary, microbial, and/or pharmacological origin, which aims at modulating indole and indole-related compounds in the biological fluids, notably in the colonic lumen, should crucially take into account the dual effects of these compounds, either beneficial or deleterious, according to the host tissues considered, and the physiological or pathophysiological context (Figure 1). For instance, given the deleterious effect of indoxyl sulfate on kidney cells in chronic kidney disease patients, it can be envisaged in that situation to either decrease the production of indole by the colonic microbiota, and/or to decrease indoxyl sulfate production by the liver, and/or to increase its removal from blood by an appropriate dialysis system.

In order to answer questions related to the effects of indolic compounds on the colonic epithelium, further research is needed to explore advantages from the utilization of colonic organoids for determining the effects of bacterial metabolites on the different phenotypes among intestinal epithelial cells [140]. Additional pre-clinical studies are needed to document further the effects of indolic compounds on host tissues, thus paving the way for future clinical studies.

## Figures and Tables

**Figure 1 microorganisms-10-00930-f001:**
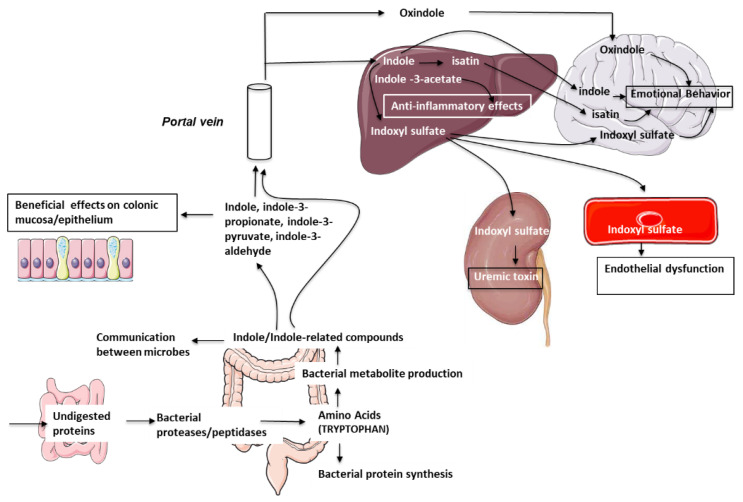
Schematic representation of the effects of indole and indole-related compounds synthesized by the intestinal microbiota on communication between microbes, and effects of bacterial metabolites and co-metabolites on host tissues. A minor portion of undigested or partially digested dietary and endogenous proteins is transferred from the small to the large intestine. There, these nitrogenous compounds are intensively degraded into their amino acid constituents by bacterial proteases and peptidases. The amino acids, including tryptophan, are used for microbial protein synthesis and bacterial metabolism, releasing numerous compounds into the luminal content. Tryptophan metabolism by specific bacterial species releases indole and several indole-related compounds. Several of these compounds are implicated in communication between intestinal microbes. The indolic compounds indole, indole-3-propionate, indole-3-pyruvate, and indole-3-aldehyde have been shown in experimental models to exert beneficial effects on the intestinal mucosa/epithelium in different physiological and pathophysiological situations. Indole and related compounds are then transferred from the lumen to the portal blood and reach the liver. In the liver, indole is partly metabolized into indoxyl sulfate. Although indole and indole-3-acetate have been shown in pre-clinical studies to exert anti-inflammatory effects on liver cells, indoxyl sulfate released from the liver has been clearly identified as a uremic toxin that is deleterious for kidney tubular cells, thus accelerating chronic kidney disease. In addition, indoxyl sulfate exerts deleterious effects on the endothelium, and is thus one element involved in endothelial dysfunction. Emerging experimental data suggest that indole, indoxyl sulfate, isatin, and oxindole are 4 indolic compounds that enter the brain by unknown mechanisms, exerting there, depending on the dose used, either beneficial or deleterious effects on brain activity and emotional behavior.

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
