# Peer review of "Production of Indole and Indole-Related Compounds by the Intestinal Microbiota and Consequences for the Host: The Good, the Bad, and the Ugly"

_microorganisms, 2022, doi:10.3390/microorganisms10050930_

Round 1

Reviewer 1 Report

This paper reviewed the advantages and disadvantages of the indole and indole-related compounds produced from L-tryptophan by the intestinal microbiota to the human health. The paper has some scientific significance, but still some improvement is required.

  1. The paper is too long and needs to be curtailed.
  2. The structure is disarranged. Some parts are based on the effects to organs or the systems. Some parts are based on the production of indole and indole-related compounds. Some effects are beneficial, some effects are harmful. I suggest reorganizing the structure to make it clearer.
  3. Some parts with more words can be subdivided into several branches to make them more logical.
  4. Indole and indole-related compounds are not the only metabolites from L-tryptophan produced by gut microbiota. Biogenic amine such as tryptamine, especially 5-HT, attracted more attention of their effects to the CNS. Although indole and indole-related compounds are the subjects of this paper. The interactions of them with other metabolites should be shortly discussed in this paper.

Author Response

1- The paper is too long and needs to be curtailed.

Response: We have curtailed the text by eliminating some redundancies and by shortening numerous sentences. We have also curtailed the conclusions (paragraph 9) by focusing on the main points of the paper.

2- The structure is disarranged. Some parts are based on the effects to organs or the systems. Some parts are based on the production of indole and indole-related compounds. Some effects are beneficial, some effects are harmful. I suggest reorganizing the structure to make it clearer.

Response: We have modified the plan of our manuscript, notably in the paragraph devoted to the effects of indole and indolic compounds on the central nervous system (part 8), in which, depending on the dose of the compounds used,  both beneficial and deleterious effects are recorded, by modifying the structure of the text and by adding subheads. We hope this will make the text easier to read.

3- Some parts with more words can be subdivided into several branches to make them more logical.

Response: We have subdivided several paragraphs, notably the longest ones, into several subsections.

4- Indole and indole-related compounds are not the only metabolites from L-tryptophan produced by gut microbiota. Biogenic amine such as tryptamine, especially 5-HT, attracted more attention of their effects to the CNS. Although indole and indole-related compounds are the subjects of this paper. The interactions of them with other metabolites should be shortly discussed in this paper.

Response: We have introduced a new short paragraph in order to present some of the interactions between indole and indole-related metabolites and other metabolites derived from tryptophan.

Reviewer 2 Report

This is a nice review on Indole and indole-related compounds produced by some bacterial species from Tryptophan. I found some minor problems with the cited references. In particular, 

  • Page 2 line 46 "Portune, Beaumont et al. 2016" not cited in the References or mistakenly identified as "Portune, K. J., A. Benitez-Paez, et al. (2017)" page 15 line 721.
  • Page 3 line 102 "Dong and Perdew 2020" is not cited in the References.

Author Response

  • Page 2 line 46 "Portune, Beaumont et al. 2016" not cited in the References or mistakenly identified as "Portune, K. J., A. Benitez-Paez, et al. (2017)" page 15 line 721.
  • Page 3 line 102 "Dong and Perdew 2020" is not cited in the References.

Response: We have made the corrections for the reference by Portune et al. and for the missing reference. Thank you for bringing this to our attention.

Reviewer 3 Report

Thank you for the chance to review your paper describing indole and its derivatives production by gut and their implications for the host.

This is an emerging and novel field (in terms of in-depth insight into gut microbiota biology) that for sure is gaining attention. 

For the most part, the paper is very solid, clearly written, and beautifully illustrated in the figure. I don't have major concerns as the paper is already a tour-de-force with a lot of important information, however, I have several suggestions that should be addressed at least in the text to increase clarity, and make fair conclusions of the data, and avoid overinterpretation. 

The impact of uremic products on cognitive functions/nervous system is pretty fascinating - I would like to see a figure showing the current state of the art in this sub-area. Since the paragraph might be hard to follow for non-experts in the field this figure will make it more clear.

I am also thinking that a brief paragraph (just to wave a flag for the problem) discussing the hardness of removal of the discussed toxins would be beneficial for the paper - it will underlie the importance of the discussed topic.

Authors pointed out the very important property of indoxyl sulfate - it might act as a trigger of thrombosis and switcher for the pro-thrombotic state when elevated in the body. I suggest additional few sentences after line 363 with a discussion of human/clinical data in this area; these papers should be handy:

https://doi.org/10.1186/s12882-017-0457-1 

https://doi.org/10.1161/JAHA.116.005022

I would definitely recommend shortening the Conclusions paragraph - it should be a concise and brief sum-up enriched with 2-3 sentences from authors showing the future directions for the area. 

Maybe introducing brief information on how binding to the proteins (albumins/globulins within the bloodstream) impacts indole/indole-derivatives biology will be beneficial.

Chapter 3 sounds terrific - this is a brand new area at a crossing of immunology/toxicology/microbiology and, if possible/doable, expanding this section will bring even more novelty to the paper.

Please check your manuscript carefully since it contains some minor spelling and grammatical errors.

To sum up, the paper is well-written and provides a piece of great novelty and fresh insights into an emerging field of medicine. I am supportive of the publication, albeit, after some major things are corrected. 

Author Response

Thank you for the chance to review your paper describing indole and its derivatives production by gut and their implications for the host.

This is an emerging and novel field (in terms of in-depth insight into gut microbiota biology) that for sure is gaining attention. 

For the most part, the paper is very solid, clearly written, and beautifully illustrated in the figure. I don't have major concerns as the paper is already a tour-de-force with a lot of important information, however, I have several suggestions that should be addressed at least in the text to increase clarity, and make fair conclusions of the data, and avoid overinterpretation. 

The impact of uremic products on cognitive functions/nervous system is pretty fascinating - I would like to see a figure showing the current state of the art in this sub-area. Since the paragraph might be hard to follow for non-experts in the field this figure will make it more clear.

Response: We have been thinking about drawing another figure, in addition to Figure 1, that would better explain the effects of indole and indole derivatives on the cognitive function/nervous system. However, taking into account that this topic is presently an emerging one, we found difficult to build such a figure. However, in order to make this part clearer, and as mentioned by referee 1, we have rewritten the paragraph 8 and added subheads in this paragraph.

I am also thinking that a brief paragraph (just to wave a flag for the problem) discussing the hardness of removal of the discussed toxins would be beneficial for the paper - it will underlie the importance of the discussed topic.

Response: We have added a new short paragraph related to the possible ways to diminish the indole-related compounds with deleterious effects. Thank you for the comment.

Authors pointed out the very important property of indoxyl sulfate - it might act as a trigger of thrombosis and switcher for the pro-thrombotic state when elevated in the body. I suggest additional few sentences after line 363 with a discussion of human/clinical data in this area; these papers should be handy:

https://doi.org/10.1186/s12882-017-0457-1 

https://doi.org/10.1161/JAHA.116.005022

Response: We have modified the paragraph related to the effects of indolic compounds on the cardiovascular functions by adding some sentences related to clinical data with the corresponding references.

I would definitely recommend shortening the Conclusions paragraph - it should be a concise and brief sum-up enriched with 2-3 sentences from authors showing the future directions for the area. 

Response: We have shortened the Conclusion paragraph in order to focus on the main points and main future directions.

Maybe introducing brief information on how binding to the proteins (albumins/globulins within the bloodstream) impacts indole/indole-derivatives biology will be beneficial.

Response: We have reinforced the part regarding the binding of indole-related compounds by proteins in plasma with the corresponding references.

Chapter 3 sounds terrific - this is a brand new area at a crossing of immunology/toxicology/microbiology and, if possible/doable, expanding this section will bring even more novelty to the paper.

Response: We have expanded the chapter 3 by giving additional information on the part played by indolic compounds on microbial community communication, together with related bibliography.

Please check your manuscript carefully since it contains some minor spelling and grammatical errors.

Response: The manuscript has been double-checked in order to eliminate spelling and grammatical errors.

To sum up, the paper is well-written and provides a piece of great novelty and fresh insights into an emerging field of medicine. I am supportive of the publication, albeit, after some major things are corrected. 

Response: Thank you for your support.

Round 2

Reviewer 3 Report

The Authors met my expectations and improved manuscript a lot. I am really satisfied with the work that has been done and now, I feel that manuscript fully fits the journal scope in terms of quality. I am happy to recommend the paper as it is for the acceptance.